# Expected Frequency Matrices of Elections: Computation, Geometry, and Preference Learning

**Niclas Boehmer**
Algorithmics and Computational Complexity
Technische Universität Berlin
niclas.boehmer@tu-berlin.de

**Robert Bredereck**
TU Clausthal
robert.bredereck@tu-clausthal.de

**Edith Elkind**
University of Oxford
elkind@cs.ox.ac.uk

**Piotr Faliszewski**
AGH University
faliszew@agh.edu.pl

**Stanisław Szufa**
AGH University
Jagiellonian University
szufa@agh.edu.pl

## Abstract

We use the "map of elections" approach of Szufa et al. (AAMAS-2020) to analyze several well-known vote distributions. For each of them, we give an explicit formula or an efficient algorithm for computing its frequency matrix, which captures the probability that a given candidate appears in a given position in a sampled vote. We use these matrices to draw the "skeleton map" of distributions, evaluate its robustness, and analyze its properties. Finally, we develop a general and unified framework for learning the distribution of real-world preferences using the frequency matrices of established vote distributions.

## 1 Introduction

Computational social choice is a research area at the intersection of social choice (the science of collective decision-making) and computer science, which focuses on the algorithmic analysis of problems related to preference aggregation and elicitation (Brandt et al., 2013). Many of the early papers in this field were primarily theoretical, focusing on establishing the worst-case complexity of winner determination and strategic behavior under various voting rules—see, e.g., the papers of Hemaspaandra et al. (1997), Dwork et al. (2001), and Conitzer et al. (2007)—but more recent work often combines theoretical investigations with empirical analysis. For example, formal bounds on the running time and/or approximation ratio of a winner determination algorithm can be complemented by experiments that evaluate its performance on realistic instances; see, e.g., the works of Conitzer (2006), Betzler et al. (2014), Faliszewski et al. (2018) and Wang et al. (2019).

However, performing high-quality experiments requires the ability to organize and understand the available data. One way to achieve this is to form a so-called "map of elections," recently introduced by Szufa et al. (2020) and extended by Boehmer et al. (2021b). The idea is as follows. First, we fix a distance measure between elections. Second, we sample a number of elections from various distributions and real-life datasets—e.g., those collected in PrefLib (Mattei & Walsh, 2013)—and measure the pairwise distances between them. Third, we embed these elections into the 2D plane, mapping each election to a point so that the Euclidean distances between points are approximately equal to the distances between the respective elections. Finally, we plot these points, usually coloring them to indicate their origin (e.g., the distribution from which a given election was sampled); see Figure 2 later in the paper for an example of such a map. A location of an election on a map provides useful information about its properties. For example, Szufa et al. (2020) and Boehmer et al. (2021a,b) have shown that it can be used to predict (a) the Borda score of the winner of the election, (b) the

36th Conference on Neural Information Processing Systems (NeurIPS 2022).

running time of ILP solvers computing the winners under the Harmonic-Borda multiwinner voting rule, or (c) the robustness of Plurality and Borda winners. Moreover, real-world elections of the same type (such as the ones from politics, sports, or surveys) tend to cluster in the same areas of the map; see also the positions on the map of the datasets collected by Boehmer & Schaar (2022). As such, the map has proven to be a useful framework to analyze the nature of elections and to visualize experimental results in a non-aggregate fashion.

Unfortunately, extending the map to incorporate additional examples and distributions is a challenging task, as the visual representation becomes cluttered and, more importantly, the embedding algorithms, which map elections to points in 2D, find it more difficult to preserve pairwise distances between points as the number of points increases. It is therefore desirable to reduce the number of points in a way that preserves the key features of the framework.

We address this challenge by drawing a map of *distributions* rather than individual elections, which we call the *skeleton map*. That is, instead of sampling 20–30 points from each distribution and placing them all on the map, as Szufa et al. (2020) and Boehmer et al. (2021b) do (obtaining around 800 points in total), we create a single point for each distribution. This approach is facilitated by the fact that prior work on the "map of elections" framework represented elections by their *frequency matrices*, which capture their essential features. The starting point of our work is the observation that this representation extends to distributions in a natural way. Thus, if we can compute the frequency matrix of some distribution $\mathcal{D}$, then, instead of sampling elections from $\mathcal{D}$ and creating a point on the map for each sample, we can create a single point for $\mathcal{D}$ itself.

**Our Contribution.** We provide three sets of results. First, for a number of prominent vote distributions, we show how to compute their frequency matrices, by providing an explicit formula or an efficient algorithm. Second, we draw the map of distributions (the *skeleton map*) and argue for its credibility and robustness. Finally, we use our results to estimate the parameters of the distributions that are closest to the real-world elections considered by Boehmer et al. (2021b). In more detail, we work in the setting of preference learning, where we are given an election and we want to learn the parameters of some distribution, so as to maximize the similarity of the votes sampled from this distribution and the input election. For example, we may be interested in fitting the classic model of Mallows (1957). This model is parameterized by a central vote $v$ and a dispersion parameter $\phi$, which specifies how likely it is to generate a vote at some distance from the central one (alternatively, one may use, e.g., the Plackett–Luce model). Previous works on preference learning typically proposed algorithms to learn the parameters of one specific (parameterized) vote distribution (see, e.g., the works of Lu & Boutilier (2014); Mandhani & Meilă (2009); Meila & Chen (2010); Vitelli et al. (2017); Murphy & Martin (2003); Awasthi et al. (2014) for (mixtures of) the Mallows model and the works of Guiver & Snelson (2009); Hunter (2004); Minka (2004); Gormley & Murphy (2008) for (mixtures of) the Plackett–Luce model). Using frequency matrices, we offer a more general approach. Indeed, given an election and a parameterized vote distribution whose frequency matrix we can compute, the task of learning the distribution's parameters boils down to finding parameters that minimize the distance between the election and the matrices of the distribution. While this minimization problem may be quite challenging, our approach offers a uniform framework for dealing with multiple kinds of distributions at the same time. We find that for the case of the Mallows distribution, our approach learns parameters very similar to those established using maximum likelihood-based approaches. Omitted proofs and discussions are in the appendix. The source code used for the experiments is available in a GitHub repository[1].

## 2 Preliminaries

Given an integer $t$, we write $[t]$ to denote the set $\{1, \ldots, t\}$. We interpret a vector $x \in \mathbb{R}^m$ as an $m \times 1$ matrix (i.e., we use column vectors as the default).

**Preference Orders and Elections.** Let $C$ be a finite, nonempty set of candidates. We refer to total orders over $C$ as *preference orders* (or, equivalently, *votes*), and denote the set of all preference orders over $C$ by $\mathcal{L}(C)$. Given a vote $v$ and a candidate $c$, by $\mathrm{pos}_v(c)$ we mean the position of $c$ in $v$ (the top-ranked candidate has position 1, the next one has position 2, and so on). If a candidate $a$ is ranked above another candidate $b$ in vote $v$, we write $v: a \succ b$. Let $\mathrm{rev}(v)$ denote the *reverse* of

---

[1]github.com/Project-PRAGMA/Expected-Frequency-Matrices-NeurIPS-2022

vote $v$. An *election* $E = (C, V)$ consists of a set $C = \{c_1, \ldots, c_m\}$ of candidates and a collection $V = (v_1, \ldots, v_n)$ of votes. Occasionally we refer to the elements of $V$ as voters rather than votes.

**Frequency Matrices.** Consider an election $E = (C, V)$ with $C = \{c_1, \ldots, c_m\}$ and $V = (v_1, \ldots, v_n)$. For each candidate $c_j$ and position $i \in [m]$, we define $\#\mathrm{freq}_E(c_j, i)$ to be the fraction of the votes from $V$ that rank $c_j$ in position $i$. We define the column vector $\#\mathrm{freq}_E(c_j)$ to be $(\#\mathrm{freq}_E(c_j, 1), \ldots, \#\mathrm{freq}_E(c_j, m))$ and matrix $\#\mathrm{freq}(E)$ to consist of vectors $\#\mathrm{freq}_E(c_1), \ldots, \#\mathrm{freq}_E(c_m)$. We refer to $\#\mathrm{freq}(E)$ as the *frequency matrix of election $E$*. Frequency matrices are bistochastic, i.e., their entries are nonnegative and each of their rows and columns sums up to one.

**Example 2.1.** Let $E = (C, V)$ be an election with candidate set $C = \{a, b, c, d, e\}$ and four voters, $v_1$, $v_2$, $v_3$, and $v_4$. Below, we show the voters' preference orders (on the left) and the election's frequency matrix (on the right).

$$
\begin{array}{ll}
v_1: a \succ b \succ c \succ d \succ e, \\
v_2: c \succ b \succ d \succ a \succ e, \\
v_3: d \succ e \succ c \succ b \succ a, \\
v_4: b \succ c \succ a \succ d \succ e.
\end{array}
\qquad
\begin{array}{c}
\phantom{1}\\
\end{array}
\begin{array}{c|ccccc}
 & a & b & c & d & e \\
1 & 1/4 & 1/4 & 1/4 & 1/4 & 0 \\
2 & 0 & 1/2 & 1/4 & 0 & 1/4 \\
3 & 1/4 & 0 & 1/2 & 1/4 & 0 \\
4 & 1/4 & 1/4 & 0 & 1/2 & 0 \\
5 & 1/4 & 0 & 0 & 0 & 3/4
\end{array}
$$

Given a vote $v$, we write $\#\mathrm{freq}(v)$ to denote the frequency matrix of the election containing this vote only; $\#\mathrm{freq}(v)$ is a permutation matrix, with a single 1 in each row and in each column. Thus, for an election $E = (C, V)$ with $V = (v_1, \ldots, v_n)$ we have $\#\mathrm{freq}(E) = \frac{1}{n} \cdot \sum_{i=1}^{n} \#\mathrm{freq}(v_i)$.

**Compass Matrices.** For even $m$, Boehmer et al. (2021b) defined the following four $m \times m$ "compass" matrices, which appear to be extreme on the "map of elections":

1. The *identity matrix*, $\mathrm{ID}_m$, has ones on the diagonal and zeroes everywhere else (it corresponds to an election where all voters agree on a single preference order).

2. The *uniformity matrix*, $\mathrm{UN}_m$, has all entries equal to $1/m$ (it corresponds to lack of agreement; each candidate is ranked at each position equally often).

3. The *stratification matrix*, $\mathrm{ST}_m$, is partitioned into four quadrangles, where all entries in the top-left and bottom-right quadrangles are equal to $2/m$, and all other entries are equal to zero (it corresponds to partial agreement; the voters agree which half of the candidates is superior, but disagree on everything else).

4. The *antagonism matrix*, $\mathrm{AN}_m$, has values $1/2$ on both diagonals and zeroes elsewhere (it captures a conflict: it is a matrix of an election where half of the voters rank the candidates in one way and half of the voters rank them in the opposite way).

Below, we show examples of these matrices for $m = 4$:

$$
\mathrm{UN}_4 = \begin{bmatrix} 1/4 & 1/4 & 1/4 & 1/4 \\ 1/4 & 1/4 & 1/4 & 1/4 \\ 1/4 & 1/4 & 1/4 & 1/4 \\ 1/4 & 1/4 & 1/4 & 1/4 \end{bmatrix}, \mathrm{ID}_4 = \begin{bmatrix} 1 & 0 & 0 & 0 \\ 0 & 1 & 0 & 0 \\ 0 & 0 & 1 & 0 \\ 0 & 0 & 0 & 1 \end{bmatrix}, \mathrm{ST}_4 = \begin{bmatrix} 1/2 & 1/2 & 0 & 0 \\ 1/2 & 1/2 & 0 & 0 \\ 0 & 0 & 1/2 & 1/2 \\ 0 & 0 & 1/2 & 1/2 \end{bmatrix}, \mathrm{AN}_4 = \begin{bmatrix} 1/2 & 0 & 0 & 1/2 \\ 0 & 1/2 & 1/2 & 0 \\ 0 & 1/2 & 1/2 & 0 \\ 1/2 & 0 & 0 & 1/2 \end{bmatrix}.
$$

We omit the subscript in the names of these matrices if its value is clear from the context or irrelevant.

**EMD.** Let $x = (x_1, \ldots, x_n)$ and $y = (y_1, \ldots, y_n)$ be two vectors with nonnegative real entries that sum up to 1. Their *Earth mover's distance*, denoted $\mathrm{EMD}(x, y)$, is the cost of transforming $x$ into $y$ using operations of the form: Given indices $i, j \in [n]$ and a positive value $\delta$ such that $x_i \geq \delta$, at the cost of $\delta \cdot |i - j|$, replace $x_i$ with $x_i - \delta$ and $x_j$ with $x_j + \delta$ (this corresponds to moving $\delta$ amount of "earth" from position $i$ to position $j$). $\mathrm{EMD}(x, y)$ can be computed in polynomial time by a standard greedy algorithm.

**Positionwise Distance (Szufa et al., 2020).** Let $A = (a_1, \ldots, a_m)$ and $B = (b_1, \ldots, b_m)$ be two $m \times m$ frequency matrices. Their *raw positionwise distance* is $\mathrm{rawPOS}(A, B) = \min_{\sigma \in S_m} \sum_{i=1}^{m} \mathrm{EMD}(a_i, b_{\sigma(i)})$, where $S_m$ denotes the set of all permutations over $[m]$. We will normalize these distances by $\frac{1}{3}(m^2 - 1)$, which Boehmer et al. (2021b, 2022) proved to be the maximum distance between two $m \times m$ frequency matrices and the distance between $\mathrm{ID}_m$ and

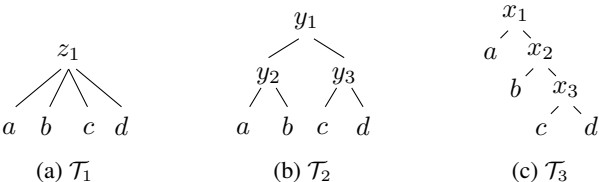

(a) $\mathcal{T}_1$        (b) $\mathcal{T}_2$        (c) $\mathcal{T}_3$

Figure 1: Three examples of clone decomposition trees.

$UN_m$: $nPOS(A, B) = \frac{rawPOS(A,B)}{\frac{1}{3}(m^2-1)}$. For two elections $E$ and $F$ with equal-sized candidate sets, their positionwise distance, raw or normalized, is defined as the positionwise distance between their frequency matrices.

**Paths Between the Compass Matrices.** Let $X$ and $Y$ be two compass matrices. Boehmer et al. (2021b) showed that if we take their affine combination $Z = \alpha X + (1 - \alpha)Y$ ($0 \leq \alpha \leq 1$) then $nPOS(X, Z) = (1 - \alpha)nPOS(X, Y)$ and $nPOS(Z, Y) = \alpha nPOS(X, Y)$. Such affine combinations form direct paths between the compass matrices; they are also possible between any two frequency matrices of a given size, not just the compass ones, but may require shuffling the matrices' columns (Boehmer et al., 2021b).

**Structured Domains.** We consider two classes of structured elections, single-peaked elections (Black, 1958), and group-separable elections (Inada, 1964). For a discussion of these domains and the motivation behind them, see the original papers and the overviews by Elkind et al. (2017, 2022).

Intuitively, an election is single-peaked if we can order the candidates so that, as each voter considers the candidates in this order (referred to as the *societal axis*), his or her appreciation first increases and then decreases. The axis may, e.g., correspond to the left-right political spectrum.

**Definition 2.2.** Let $v$ be a vote over $C$ and let $\lhd$ be the societal axis over $C$. We say that $v$ is *single-peaked with respect to* $\lhd$ if for every $t \in [|C|]$ its $t$ top-ranked candidates form an interval within $\lhd$. An election is *single-peaked with respect to* $\lhd$ if all its votes are. An election is *single-peaked (SP)* if it is single-peaked with respect to some axis.

Note that the election from Example 2.1 is single-peaked with respect to the axis $a \lhd b \lhd c \lhd d \lhd e$.

We also consider *group-separable elections*, introduced by Inada (1964). For our purposes, it will be convenient to use the tree-based definition of Karpov (2019). Let $C = \{c_1, \ldots, c_m\}$ be a set of candidates, and consider a rooted, ordered tree $\mathcal{T}$ whose leaves are elements of $C$. The *frontier* of such a tree is the preference order that ranks the candidates in the order in which they appear in the tree from left to right. A preference order is *consistent* with a given tree if it can be obtained as its frontier by reversing the order in which the children of some nodes appear.

**Definition 2.3.** An election $E = (C, V)$ is *group-separable* if there is a rooted, ordered tree $\mathcal{T}$ whose leaves are members of $C$, such that each vote in $V$ is consistent with $\mathcal{T}$.

The trees from Definition 2.3 form a subclass of *clone decomposition trees*, which are examples of PQ-trees (Elkind et al., 2012; Booth & Lueker, 1976).

**Example 2.4.** Consider candidate set $C = \{a, b, c, d\}$, trees $\mathcal{T}_1$, $\mathcal{T}_2$, and $\mathcal{T}_3$ from Figure 1, and votes $v_1 \colon a \succ b \succ c \succ d$, $v_2 \colon c \succ d \succ b \succ a$, and $v_3 \colon b \succ d \succ c \succ a$. Vote $v_1$ is consistent with each of the trees, $v_2$ is consistent with $\mathcal{T}_2$ (reverse the children of $y_1$ and $y_2$), and $v_3$ is consistent with $\mathcal{T}_3$ (reverse the children of $x_1$ and $x_3$).

## 3 Frequency Matrices for Vote Distributions

We show how to compute frequency matrices for several well-known distributions over votes.

### 3.1 Setup and Interpretation

A *vote distribution* for a candidate set $C$ is a function $\mathcal{D}$ that assigns a probability to each preference order over $C$. Formally, we require that for each $v \in \mathcal{L}(C)$ it holds that $\mathcal{D}(v) \geq 0$ and

$\sum_{v \in \mathcal{L}(C)} \mathcal{D}(v) = 1$. We say that a vote $v$ is in the *support* of $\mathcal{D}$ if $\mathcal{D}(v) > 0$. Given such a distribution, we can form an election by repeatedly drawing votes according to the specified probabilities. For example, we can sample each element of $\mathcal{L}(C)$ with equal probability; this distribution, which is known as *impartial culture (IC)*, is denoted by $\mathcal{D}_{\text{IC}}$ (we omit the candidate set from our notation as it will always be clear from the context). The *frequency matrix of a vote distribution $\mathcal{D}$* over a candidate set $C$ is $\#\text{freq}(\mathcal{D}) = \sum_{v \in \mathcal{L}(C)} \mathcal{D}(v) \cdot \#\text{freq}(v)$. For example, we have $\#\text{freq}(\mathcal{D}_{\text{IC}}) = \text{UN}$. One interpretation of $\#\text{freq}(\mathcal{D})$ is that the entry for a candidate $c_j$ and a position $i$ is the probability that a vote $v$ sampled from $\mathcal{D}$ has $c_j$ in position $i$ (which we denote as $\mathbb{P}[\text{pos}_v(c_j) = i]$). Another interpretation is that if we sample a large number of votes then the resulting election's frequency matrix would be close to $\#\text{freq}(\mathcal{D})$ with high probability. More formally, if we let $\mathcal{M}_n$ be a random variable equal to the frequency matrix of an $n$-voter election generated according to $\mathcal{D}$, then it holds that $\lim_{n \to \infty} \mathbb{E}(\mathcal{M}_n) = \#\text{freq}(\mathcal{D})$.

## 3.2 Group-Separable Elections

We first consider sampling group-separable votes. Given a rooted tree $\mathcal{T}$ whose leaves are labeled by elements of $C = \{c_1, \ldots, c_m\}$, let $\mathcal{D}_{\text{GS}}^{\mathcal{T}}$ be the distribution assigning equal probability to all votes consistent with $\mathcal{T}$, and zero probability to all other votes; one can think of $\mathcal{D}_{\text{GS}}^{\mathcal{T}}$ as impartial culture restricted to the group-separable subdomain defined by $\mathcal{T}$. To sample from $\mathcal{D}_{\text{GS}}^{\mathcal{T}}$, we can toss a fair coin for each internal node of $\mathcal{T}$, reversing the order of its children if this coin comes up heads, and output the frontier of the resulting tree. We focus on the following types of trees:

1. $\text{Flat}(c_1, \ldots, c_m)$ is a tree with a single internal node, whose children, from left to right, are $c_1, c_2, \ldots, c_m$. There are only two preference orders consistent with this tree, $c_1 \succ \cdots \succ c_m$ and its reverse.

2. $\text{Bal}(c_1, \ldots, c_m)$ is a perfectly balanced binary tree with frontier $c_1, \ldots, c_m$ (hence we assume the number $m$ of candidates to be a power of two).

3. $\text{CP}(c_1, \ldots, c_m)$ is a binary caterpillar tree: it has internal nodes $x_1, \ldots, x_{m-1}$; for each $j \in [m-2]$, $x_j$ has $c_j$ as the left child and $x_{j+1}$ as the right one, whereas $x_{m-1}$ has both $c_{m-1}$ and $c_m$ as children.

The first tree in Figure 1 is flat, the second one is balanced, and the third one is a caterpillar tree. If $\mathcal{T}$ is a caterpillar tree, then we refer to $\mathcal{D}_{\text{GS}}^{\mathcal{T}}$ as the *GS/caterpillar distribution*. We use a similar terminology for the other trees.

**Theorem 3.1.** *Let $F$ be the frequency matrix of distribution $\mathcal{D}_{\text{GS}}^{\mathcal{T}}$. If $\mathcal{T}$ is flat then $F = \text{AN}$, and if it is balanced then $F = \text{UN}$. If $\mathcal{T}$ is a caterpillar tree $\text{CP}(c_1, \ldots, c_m)$, then for each candidate $c_j$ the probability that $c_j$ appears in a position $i \in [m]$ in a random vote $v$ sampled from $\mathcal{D}_{\text{GS}}^{\mathcal{T}}$ is:*

$$\frac{1}{2^j} \binom{j-1}{i-1} \cdot \mathbb{1}_{i \leq j} + \frac{1}{2^j} \binom{j-1}{(i-1)-(m-j)} \cdot \mathbb{1}_{i > m-j}.$$

*Proof.* The cases of flat and balanced trees are immediate, so we focus on caterpillar trees. Let $\mathcal{T} = \text{CP}(c_1, \ldots, c_m)$ with internal nodes $x_1, \ldots, x_{m-1}$, and consider a candidate $c_j$ and a position $i \in [m]$. Let $v$ be a random variable equal to a vote sampled from $\mathcal{D}_{\text{GS}}^{\mathcal{T}}$. We say that a node $x_\ell$, $\ell \in [m-1]$, is *reversed* if the order of its children is reversed. Note that for $\ell < r$ it holds that $c_r$ precedes $c_\ell$ in the frontier if and only if $x_\ell$ is reversed. Suppose that $x_j$ is not reversed. Then $v$ ranks $c_j$ above each of $c_{j+1}, \ldots, c_m$. This means that for $c_j$ to be ranked exactly in position $i$, it must be that $j \geq i$ and exactly $i-1$ nodes among $x_1, \ldots, x_{j-1}$ are not reversed. If $j \geq i$, the probability that $x_j$ and $i-1$ nodes among $x_1, \ldots, x_{j-1}$ are not reversed is $\frac{1}{2^j} \cdot \binom{j-1}{i-1}$. On the other hand, if $x_j$ is reversed, then $v$ ranks candidates $c_{j+1}, \ldots, c_m$ above $c_j$. As there are $m-j$ of them, for $c_j$ to be ranked exactly in position $i$ it must hold that $i > m-j$ and exactly $(i-1)-(m-j)$ nodes among $x_1, \ldots, x_{j-1}$ are not reversed. This happens with probability $\frac{1}{2^j} \cdot \binom{j-1}{(i-1)-(m-j)}$. $\qquad\square$

Regarding distributions $\mathcal{D}_{\text{GS}}^{\mathcal{T}}$ not handled in Theorem 3.1, we still can compute their frequence matrices efficiently.

**Theorem 3.2.** *There is an algorithm that given a tree $\mathcal{T}$ computes $\#\text{freq}(D_{\text{GS}}^{\mathcal{T}})$ using polynomially many arithmetic operations with respect to the number of nodes in $\mathcal{T}$.*

## 3.3 From Caterpillars to Single-Peaked Preferences.

There is a relationship between GS/caterpillar votes and single-peaked ones, which will be very useful when computing one of the frequency matrices in the next section.

**Theorem 3.3.** *Given a ranking $v$ over $C = \{c_1, \ldots, c_m\}$, let $\widehat{v}$ be another ranking over $C$ such that, for each $j \in [m]$, if $c_j$ is ranked in position $i$ in $v$ then $c_i$ is ranked in position $m - j + 1$ in $\widehat{v}$. Suppose that $v$ is in the support of $\mathcal{D}_{\mathrm{GS}}^{\mathcal{T}}$, where $\mathcal{T} = \mathrm{CP}(c_1, \ldots, c_m)$. Then $\widehat{v}$ is single-peaked with respect to $c_1 \lhd \cdots \lhd c_m$.*

There are exactly $2^{m-1}$ votes in the support of $\mathcal{D}_{\mathrm{GS}}^{\mathcal{T}}$ (this follows by simple counting) and there are $2^{m-1}$ votes that are single-peaked with respect to $c_1 \lhd \cdots \lhd c_m$. As $u \neq v$ implies $\widehat{u} \neq \widehat{v}$, it follows that the mapping $v \mapsto \widehat{v}$ is a bijection between all votes in the support of $\mathcal{D}_{\mathrm{GS}}^{\mathcal{T}}$ and all votes that are single-peaked with respect to $c_1 \lhd \cdots \lhd c_m$.

## 3.4 Single-Peaked Elections

We consider two models of generating single-peaked elections, one due to Walsh (2015) and one due to Conitzer (2009). Let us fix a candidate set $C = \{c_1, \ldots, c_m\}$ and a societal axis $c_1 \lhd \cdots \lhd c_m$. Under the Walsh distribution, denoted $\mathcal{D}_{\mathrm{SP}}^{\mathrm{Wal}}$, each vote that is single-peaked with respect to $\lhd$ has equal probability (namely, $\frac{1}{2^{m-1}}$), and all other votes have probability zero. By Theorems 3.1 and 3.3, we immediately obtain the frequency matrix for the Walsh distribution (in short, it is the transposed matrix of the GS/caterpillar distribution).

**Corollary 3.4.** *Consider a candidate set $C = \{c_1, \ldots, c_m\}$ and an axis $c_1 \lhd \cdots \lhd c_m$. The probability that candidate $c_j$ appears in position $i$ in a vote sampled from $\mathcal{D}_{\mathrm{SP}}^{\mathrm{Wal}}$ is: $\frac{1}{2^{m-i+1}} \binom{m-i}{j-1} \cdot \mathbb{1}_{j \leq m-i+1} + \frac{1}{2^{m-i+1}} \binom{m-i}{j-i} \cdot \mathbb{1}_{j > i-1}$.*

To sample a vote from the Conitzer distribution, $\mathcal{D}_{\mathrm{SP}}^{\mathrm{Con}}$ (also known as the *random peak distribution*), we pick some candidate $c_j$ uniformly at random and rank him or her on top. Then we perform $m - 1$ iterations, where in each we choose (uniformly at random) a candidate directly to the right or the left of the already selected ones, and place him or her in the highest available position in the vote.

**Theorem 3.5.** *Let $c_1 \lhd \cdots \lhd c_m$ be the societal axis, where $m$ is an even number, and let $v$ be a random vote sampled from $\mathcal{D}_{\mathrm{SP}}^{\mathrm{Con}}$ for this axis. For $j \in [\frac{m}{2}]$ and $i \in [m]$ we have:*

$$
\mathbb{P}[\mathrm{pos}_v(c_j) = i] = \begin{cases} 2/2m & \text{if } i < j, \\ (j+1)/2m & \text{if } i = j, \\ 1/2m & \text{if } j < i < m - j + 1, \\ (m-j+1)/2m & \text{if } i = m - j + 1, \\ 0 & \text{if } i + j > m. \end{cases}
$$

*Further, for each candidate $c_j \in C$ and each position $i \in [m]$ we have $\mathbb{P}[\mathrm{pos}_v(c_j) = i] = \mathbb{P}[\mathrm{pos}_v(c_{m-j+1}) = i]$.*

## 3.5 Mallows Model

Finally, we consider the classic Mallows distribution. It has two parameters, a central vote $v^*$ over $m$ candidates and a dispersion parameter $\phi \in [0, 1]$. The probability of sampling a vote $v$ from this distribution (denoted $\mathcal{D}_{\mathrm{Mal}}^{v^*, \phi}$) is: $\mathcal{D}_{\mathrm{Mal}}^{v^*, \phi}(v) = \frac{1}{Z} \phi^{\kappa(v, v^*)}$, where $Z = 1 \cdot (1 + \phi) \cdot (1 + \phi + \phi^2) \cdots \cdots (1 + \cdots + \phi^{m-1})$ is a normalizing constant and $\kappa(v, v^*)$ is the swap distance between $v$ and $v^*$ (i.e., the number of swaps of adjacent candidates needed to transform $v$ into $v^*$). In our experiments, we consider a new parameterization, introduced by Boehmer et al. (2021b). It uses a *normalized dispersion parameter* norm-$\phi$, which is converted to a value of $\phi$ so that the expected swap distance between the central vote $v^*$ and a sampled vote $v$ is $\frac{\mathrm{norm}\text{-}\phi}{2}$ times the maximum swap distance between two votes (so, norm-$\phi = 1$ is equivalent to IC and for norm-$\phi = 0.5$ we get elections that lie close to the middle of the UN–ID path).

Our goal is now to compute the frequency matrix of $\mathcal{D}_{\mathrm{Mal}}^{v^*, \phi}$. That is, given the candidate ranked in position $j$ in the central vote, we want to compute the probability that he or she appears in a

given position $i \in [m]$ in the sampled vote. Given a positive integer $m$, consider the candidate set $C(m) = \{c_1, \ldots, c_m\}$ and the central vote $v_m^*: c_1 \succ \cdots \succ c_m$. Fix a candidate $c_j \in C(m)$, and a position $i \in [m]$. For every integer $k$ between 0 and $m(m-1)/2$, let $S(m, k)$ be the number of votes in $\mathcal{L}(C(m))$ that are at swap distance $k$ from $v_m^*$, and define $T(m, k, j, i)$ to be the number of such votes that have $c_j$ in position $i$. One can compute $S(m, k)$ in time polynomial in $m$ (OEIS Foundation Inc., 2020); using $S(m, k)$, we show that the same holds for $T(m, k, j, i)$.

**Lemma 3.6.** *There is an algorithm that computes $T(m, k, j, i)$ in polynomial time with respect to $m$.*

We can now express the probability of sampling a vote $v$, where the candidate ranked in position $j$ in the central vote $v^*$ ends up in position $i$ under $\mathcal{D}_{\text{Mal}}^{v^*, \phi}$, as:

$$f_m(\phi, j, i) = \tfrac{1}{Z} \sum_{k=0}^{m(m-1)/2} T(m, k, j, i)\phi^k. \tag{1}$$

The correctness follows from the definitions of $T$ and $\mathcal{D}_{\text{Mal}}^{v^*, \phi}$. By Lemma 3.6, we have the following.

**Theorem 3.7.** *There exists an algorithm that, given a number $m$ of candidates, a vote $v^*$, and a parameter $\phi$, computes the frequency matrix of $\mathcal{D}_{\text{Mal}}^{v^*, \phi}$ using polynomially many operations in $m$.*

Note that Equation (1) only depends on $\phi$, $j$ and $i$ (and, naturally, on $m$). Using this fact, we can also compute frequency matrices for several variants of the Mallows distribution.

**Remark 3.8.** Given a vote $v$, two dispersion parameters $\phi$ and $\psi$, and a probability $p \in [0, 1]$, we define the distribution $p\text{-}\mathcal{D}_{\text{Mal}}^{v,\phi,\psi}$ as $p \cdot \mathcal{D}_{\text{Mal}}^{v,\phi} + (1-p) \cdot \mathcal{D}_{\text{Mal}}^{\text{rev}(v),\psi}$, i.e., with probability $p$ we sample a vote from $\mathcal{D}_{\text{Mal}}^{v,\phi}$ and with probability $1-p$ we sample a vote from $\mathcal{D}_{\text{Mal}}^{\text{rev}(v),\psi}$. The probability that candidate $c_j$ appears in position $i$ in the resulting vote is $p \cdot f_m(\phi, j, i) + (1-p) \cdot f_m(\psi, m-j+1, i)$.

**Remark 3.9.** Consider a candidate set $C = \{c_1, \ldots, c_m\}$. Given a vote distribution $\mathcal{D}$ over $\mathcal{L}(C)$ and a parameter $\phi$, define a new distribution $\mathcal{D}'$ as follows: Draw a vote $\hat{v}$ according to $\mathcal{D}$ and then output a vote $v$ sampled from $\mathcal{D}_{\text{Mal}}^{\hat{v},\phi}$; indeed, such models are quite natural, see, e.g., the work of Kenig & Kimelfeld (2019). For each $t \in [m]$, let $g(j, t)$ be the probability that $c_j$ appears in position $t$ in a vote sampled from $\mathcal{D}$. The probability that $c_j$ appears in position $i \in [m]$ in a vote sampled from $\mathcal{D}'$ is $\sum_{t=1}^{m} g(j, t) \cdot f(\phi, t, i)$. In terms of matrix multiplication, this means that $\#\text{freq}(\mathcal{D}') = \#\text{freq}(\mathcal{D}_{\text{Mal}}^{v^*,\phi}) \cdot \#\text{freq}(\mathcal{D})$, where $v^*$ is $c_1 \succ \cdots \succ c_m$. We write $\phi$-Conitzer ($\phi$-Walsh) to refer to this model where we use the Conitzer (Walsh) distribution as the underlying one and normalized dispersion parameter $\phi$.

# 4 Skeleton Map

Our goal in this section is to form what we call a *skeleton map of vote distributions* (skeleton map, for short), evaluate its quality and robustness, and compare it to the map of Boehmer et al. (2021b). Throughout this section, whenever we speak of a distance between elections or matrices, we mean the positionwise distance (occasionally we will also refer to the Euclidean distances on our maps, but we will always make this explicit). Let $\Phi = \{0, 0.05, 0.1, \ldots, 1\}$ be a set of normalized dispersion parameters that we will be using for Mallows-based distributions in this section.

We form the skeleton map following the general approach of Szufa et al. (2020) and Boehmer et al. (2021b). For a given number of candidates, we consider the four compass matrices (UN, ID, AN, ST) and paths between each matrix pair consisting of their convex combinations (gray dots), the frequency matrices of the Mallows distribution with normalized dispersion parameters from $\Phi$ (blue triangles), and the frequency matrices of the Conitzer (CON), Walsh (WAL), and GS/caterpillar distribution (CAT). Moreover, we add the frequency matrices of the following vote distributions (we again use the dispersion parameters from $\Phi$): (i) The distribution $1/2\text{-}\mathcal{D}_{\text{Mal}}^{v,\phi,\phi}$ as defined in Remark 3.8 (red triangles), (ii) the distribution where with equal probability we mix the standard Mallows distribution and $1/2\text{-}\mathcal{D}_{\text{Mal}}^{v,\phi,\phi}$ (green triangles), and (iii) the $\phi$-Conitzer and $\phi$-Walsh distributions as defined in Remark 3.9 (magenta and orange crosses). For each pair of these matrices we compute their positionwise distance. Then we find an embedding of the matrices into a 2D plane, so that each matrix is a point and the Euclidean distances between these points are as similar to the positionwise distances as possible (we use the MDS algorithm, as implemented in the Python sklearn.manifold.MDS package). In Figure 3 we show our map for the case of 10 candidates (the

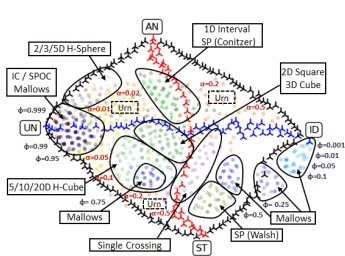
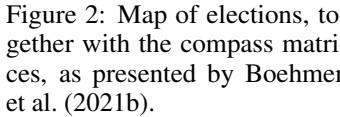

Figure 2: Map of elections, together with the compass matrices, as presented by Boehmer et al. (2021b).

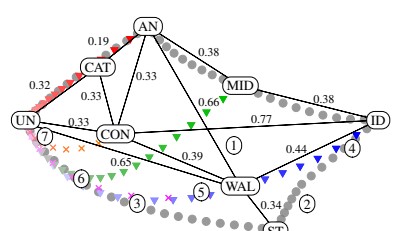

Figure 3: The skeleton map with 10 candidates. We have MID = $^1/_2$AN + $^1/_2$ID. Each point labeled with a number is a real-world election as described in Section 5.

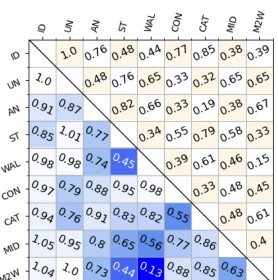

Figure 4: In the top-right part, we show the normalized positionwise distance. In the bottom-left one, we show the embedding misrepresentation ratios.

lines between some points/matrices show their positionwise distances; to maintain clarity, we only provide some of them).

We now verify the credibility of the skeleton map. As the map does not have many points, we expect its embedding to truly reflect the positionwise distances between the matrices. This, indeed, seems to be the case, although some distances are represented (much) more accurately than the others. In Figure 4 we provide the following data for a number of matrices (for $m = 10$; matrix M2W is the Mallows matrix in our data set that is closest to the Walsh matrix). In the top-right part (the white-orange area), we give positionwise distances between the matrices, and in the bottom-left part (the blue area), for each pair of matrices $X$ and $Y$ we report the *misrepresentation ratio* $\frac{\text{Euc}(X,Y)}{\text{nPOS}(X,Y)}$, where $\text{Euc}(X, Y)$ is the Euclidean distance between $X$ and $Y$ in the embedding, normalized by the Euclidean distance between ID and UN. The closer they are to 1, the more accurate is the embedding. The misrepresentation ratios are typically between 0.8 and 1.15, with many of them between 0.9 and 1.05. Thus, in most cases, the map is quite accurate and offers good intuition about the relations between the matrices. Yet, some distances are represented particularly badly. As an extreme example, the Euclidean distance between the Walsh matrix and the closest Mallows matrix, M2W, is off by almost a factor of 8 (these matrices are close, but not as close as the map suggests). Thus, while one always has to verify claims suggested by the skeleton map, we view it as quite credible. This conclusion is particularly valuable when we compare the skeleton map and the map of Boehmer et al. (2021b), shown in Figure 2. The two maps are similar, and analogous points (mostly) appear in analogous positions. Perhaps the biggest difference is the location of the Conitzer matrix on the skeleton map and Conitzer elections in the map of Boehmer et al., but even this difference is not huge. We remark that the Conitzer matrix is closer to UN and AN than to ID and ST, whereas for the Walsh matrix the opposite is true. Boehmer et al. (2021b) make a similar observation; our results allow us to make this claim formal. In Appendix E, we analyze the robustness of the skeleton map with respect to varying the number of candidates. We find that except for pairs including the Walsh or GS/caterpillar matrices, which "travel" on the map as the number of candidates increases, the distance between each pair of matrices in the skeleton map stays nearly constant.

## 5 Learning Vote Distributions

We demonstrate how the positionwise distance and frequency matrices can be used to fit vote distributions to given real-world elections. Specifically, we consider the Mallows model ($\mathcal{D}_{\text{Mal}}^{v,\phi}$) and the $\phi$-Conitzer and $\phi$-Walsh models. Naturally, we could use more distributions, but we focus on showcasing the technique and the general unified approach. Concerning our results, among others, we verify that for Mallows model our approach is strongly correlated with existing maximum-likelihood approaches. Moreover, unlike in previous works, we compare the capabilities of different distributions to fit the given elections. We remark that if we do not have an algorithm for computing a frequency matrix of a given vote distribution, we can obtain an approximate matrix by sampling sufficiently many votes from this distribution. In principle, it is also possible to deal with distributions

Table 1: Closest distributions for seven illustrative real-world elections. For each, we include the parameterization that produces the closest frequency matrix, and give the normalized positionwise distance of the elections from this matrix.

| id | source | $\mathcal{D}_{\mathrm{Mal}}^{v,\phi}$ | | $\phi$-Walsh | | $\phi$-Conitzer | |
|---|---|---|---|---|---|---|---|
| | | $\phi$ | dist | $\phi$ | dist | $\phi$ | dist |
| 1 | f. skate | 0.44 | 0.32 | 0.14 | 0.38 | 0.11 | 0.44 |
| 2 | f. skate | 0.23 | 0.24 | 0 | 0.36 | 0.02 | 0.56 |
| 3 | aspen | 0.68 | 0.18 | 0.3 | 0.16 | 0.23 | 0.27 |
| 4 | f. skate | 0.05 | 0.09 | 0 | 0.37 | 0 | 0.69 |
| 5 | s. skate | 0.46 | 0.11 | 0.15 | 0.18 | 0.16 | 0.35 |
| 6 | irish | 0.75 | 0.1 | 0.42 | 0.12 | 0.36 | 0.16 |
| 7 | cities | 0.93 | 0.06 | 0.69 | 0.06 | 0.63 | 0.06 |

over elections that do not correspond to vote distributions and hence are not captured by expected frequency matrices (as is the case, e.g., for the Euclidean models where candidates do not have fixed positions; see the work of Szufa et al. (2020) for examples of such models in the context of the map of elections): If we want to compute the distance of such a distribution, we sample sufficiently many elections and compute their average distance from the input one. However, it remains unclear how robust this approach is.

**Approach.** To fit our vote distributions to a given election, we compute the election's distance to the frequency matrices of $\mathcal{D}_{\mathrm{Mal}}^{v,\phi}$, $\phi$-Conitzer, and $\phi$-Walsh, for $\phi \in \{0, 0.001, \ldots, 1\}$. We select the distribution corresponding to the closest matrix.

**Data.** We consider elections from the real-world datasets used by Boehmer et al. (2021b). They generated 15 elections with 10 candidates and 100 voters (with strict preferences) from each of the eleven different real-world election datasets (so, altogether, they generated 165 elections, most of them from Preflib (Mattei & Walsh, 2013)). They used four datasets of political elections (from North Dublin (Irish), various non-profit and professional organizations (ERS), and city council elections from Glasgow and Aspen), four datasets of sport-based elections (from Tour de France (TDF), Giro d'Italia (GDI), speed skating, and figure skating) and three datasets with survey-based elections (from preferences over T-shirt designs, sushi, and cities). We present the results of our analysis for seven illustrative and particularly interesting elections in Table 1 and also include them in our skeleton map from Figure 3.

**Basic Test.** There is a standard maximum-likelihood estimator (MLE; based on Kemeny voting (Mandhani & Meilă, 2009)) that given an election provides the most likely dispersion parameter of the Mallows distribution that might have generated this election. To test our approach, we compared the parameters provided by our approach and by the MLE for our 165 elections and found them to be highly correlated (with Pearson correlation coefficient around 0.97). In particular, the average absolute difference between the dispersion parameter calculated by our approach and the MLE is only 0.02. See Appendix F for details.

**Fitting Real-World Elections.** Next, we consider the capabilities of $\mathcal{D}_{\mathrm{Mal}}^{v,\phi}$, $\phi$-Conitzer, and $\phi$-Walsh to fit the real-world elections of Boehmer et al. (2021b). Overall, we find that these three vote distributions have some ability to capture the considered elections, but it certainly is not perfect. Indeed, the average normalized distance of these elections to the frequency matrix of the closest distribution is 0.14. To illustrate that some distance is to be expected here, we mention that the average distance of an election sampled from impartial culture ($\mathcal{D}_{\mathrm{IC}}$, with 10 candidates and 100 voters) to the distribution's expected frequency matrix is 0.09 (see Appendix E.4 for a discussion of this and how it may serve as an estimator for the "variance of a distribution"). There are also some elections that are not captured by any of the considered distributions to an acceptable degree; examples of this are elections nr. 1 and nr. 2, which are at distance at least 0.32 and 0.25 from all our distributions, respectively. Remarkably, while coming from the same dataset, elections nr. 1 and nr. 2 are still quite different from each other and, accordingly, the computed dispersion parameter is also quite different. It remains a challenge to find distributions capturing such elections.

Comparing the power of the three considered models, nearly all of our elections are best captured by the Mallows model rather than $\phi$-Conitzer or $\phi$-Walsh. There are only twenty elections that are closer to $\phi$-Walsh or $\phi$-Conitzer than to a Mallows model (election nr. 3 is the most extreme example), and, unsurprisingly, both $\phi$-Walsh and $\phi$-Conitzer perform particularly badly at capturing elections close to ID (see election nr. 4). That is, $\phi$-Conitzer and $\phi$-Walsh are not needed to ensure good coverage of the space of elections; the average normalized distance of our elections to the closest Mallows model is only 0.0007 higher than their distance to the closest distribution (elections nr. 3-6 are three examples of elections which are well captured by the Mallows model and distributed over the entire map).[2] Nevertheless, $\phi$-Walsh is also surprisingly powerful, as the average normalized distance of our elections to the closest $\phi$-Walsh distribution is only 0.03 higher than their distance to the closest distribution (however, this might be also due to the fact that most of the considered real-world elections fall into the same area of the map, which $\phi$-Walsh happens to capture particularly well (Boehmer et al., 2021b)). $\phi$-Conitzer performs considerably worse: there are only three elections for which it produces a (slightly) better result than $\phi$-Walsh.

Moreover, our results also emphasize the complex nature of the space of elections: Election nr. 7 is very close to $\mathcal{D}_{\mathrm{Mal}}^{v,0.95}$, hinting that its votes are quite chaotic. At the same time, this election is very close to 0.63-Conitzer and 0.69-Walsh distributions, which suggests at least a certain level of structure among its votes (because votes from Conitzer and Walsh distributions are very structured, and the Mallows filter with dispersion between 0.63 and 0.69 does not destroy this structure fully). However, as witnessed by the fact that the frequency matrix of GS/balanced (which is highly structured) is UN, such phenomena can happen. Lastly, note that most of our datasets are quite "homogenous", in that the closest distributions for elections from the dataset are similar and also at a similar distance. However, there are also clear exceptions, for instance, elections nr. 1 and nr. 4 from the figure skating dataset. Moreover, there are two elections from the speed skating dataset where one election is captured best by $\mathcal{D}_{\mathrm{Mal}}^{v,0.76}$ and the other by $\mathcal{D}_{\mathrm{Mal}}^{v,0.32}$.

# 6 Summary

We have computed the frequency matrices (Szufa et al., 2020; Boehmer et al., 2021b) of several well-known distributions of votes. Using them, we have drawn a "skeleton map", which shows how these distributions relate to each other, and we have analyzed its properties. Moreover, we have demonstrated how our results can be used to fit vote distributions to capture real-world elections.

For future work, it would be interesting to compute the frequency matrices of further popular vote distributions, such as the Plackett–Luce model (we conjecture that its frequency matrix is computable in polynomial time). It would also be interesting to use our approach to fit more complex models, such as mixtures of Mallows models, to real-world elections. Further, it may be interesting to use expected frequency matrices to reason about the asymptotic behavior of our models. For example, it might be possible to formally show where, in the limit, do the matrices of our models end up on the map as we increase the number of candidates.

**Acknowledgments**

NB was supported by the DFG project MaMu (NI 369/19) and by the DFG project ComSoc-MPMS (NI 369/22). This project has received funding from the European Research Council (ERC) under the European Union's Horizon 2020 research and innovation programme (grant agreement No 101002854).

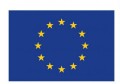 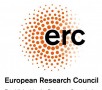

---

[2]For each election, we also computed the closest frequency matrix of two mixtures of Mallows models with reversed central votes $p$-$\mathcal{D}_{\mathrm{Mal}}^{v,\phi,\psi}$ using our approach. However, this only decreased the average minimum distance by around 0.02, with the probability $p$ of flipping the central vote being (close to) zero for most elections.

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
