# A Missing Material From Section 3.2

**Theorem 3.2.** *There is an algorithm that given a tree $\mathcal{T}$ computes $\#\mathrm{freq}(D_{\mathrm{GS}}^{\mathcal{T}})$ using polynomially many arithmetic operations with respect to the number of nodes in $\mathcal{T}$.*

*Proof.* Let $\mathcal{T}$ be the input tree, and let $C = \{c_1, \ldots, c_m\}$ be its candidate set. We will give an algorithm for computing the probability that a given candidate $c_j$ appears in position $i \in [m]$ in a vote sampled from $\mathcal{D}_{\mathrm{GS}}^{\mathcal{T}}$.

Let $x$ be some node of $\mathcal{T}$ (either internal or a leaf). Let $\mathcal{T}_x$ be the tree obtained from $\mathcal{T}$ by deleting all descendants of $x$, so that $x$ becomes a leaf, and for each subset $S$ of internal nodes of $\mathcal{T}_x$, let $\mathcal{T}_x^S$ be the ordered tree obtained by starting with $\mathcal{T}_x$ and reversing the nodes in the set $S$. For each $t \in \{0\} \cup [m-1]$ we define $f(x, t)$ to be the probability that if we reversed each internal node of $\mathcal{T}_x$ with probability $1/2$ then $x$ would be preceded by exactly $t$ candidates in the frontier of the resulting ordered tree. We compute $f(x, t)$ using dynamic programming.

Let $root$ be the root of $\mathcal{T}$. Then $f(root, 0) = 1$, and $f(root, t) = 0$ for $t \in [1, m-1]$. Next, let $x$ be some node of $\mathcal{T}$ other than the root, let $p$ be the parent of $x$, and let $\ell$ and $r$ be the number of leaves that are descendants of $x$'s left siblings and $x$'s right siblings in $\mathcal{T}$, respectively. We claim that for each $t \in \{0\} \cup [m-1]$ we have:

$$f(x, t) = \tfrac{1}{2}f(p, t - \ell) + \tfrac{1}{2}f(p, t - r).$$

To see why this formula is correct, observe that if $p \notin S$ then $x$ appears in position $t$ in the frontier of $\mathcal{T}_x^S$ if and only if $p$ appears in position $t - \ell$ in the frontier of $\mathcal{T}_p^S$: indeed, in the frontier of $\mathcal{T}_x^S$ the node $x$ appears after all predecessors of $p$ in the frontier of $\mathcal{T}_p^S$ as well as after the $\ell$ leaves that are the descendants of $x$'s left siblings in $\mathcal{T}$. Similarly, if $p \in S$ then $x$ appears in position $t$ in the frontier of $\mathcal{T}_x^S$ if and only if $p$ appears in position $t - r$ in the frontier of $\mathcal{T}_p^{S \setminus \{p\}}$: indeed, in the frontier of $\mathcal{T}_x^S$ the node $x$ appears after all predecessors of $p$ in the frontier of $\mathcal{T}_p^{S \setminus \{p\}}$ as well as after the $r$ leaves that are the descendants of $x$'s right siblings in $\mathcal{T}$. Since $p$ is reversed with probability $\frac{1}{2}$, the recurrence follows.

The above formula and standard dynamic programming allow us to compute all the values of $f$ using $O(m^2)$ arithmetic operations (note that there are at most $O(m)$ internal nodes). To complete the proof, observe that the probability that candidate $c_j$ ends up in position $i$ is $f(c_j, i - 1)$. □

# B Missing Material From Section 3.3

We use the following lemma, which is implicit in the work of Faliszewski et al. (2020).

**Lemma B.1.** *Let $\mathcal{T} = \mathrm{CP}(c_1, \ldots, c_m)$. A ranking $v$ over $\{c_1, \ldots, c_m\}$ belongs to the support of $\mathcal{D}_{\mathrm{GS}}^{\mathcal{T}}$ if and only if there exists a subset $C' \subseteq C \setminus \{c_m\}$ such that in $v$:*

1. *(1) all alternatives in $C'$ are ranked above $c_m$ and all alternatives in $(C \setminus \{c_m\}) \setminus C'$ are ranked below $c_m$;*

2. *(2) for all $c_i, c_j \in C'$ with $i < j$ the alternative $c_i$ is ranked above $c_j$;*

3. *(3) for all $c_i, c_j \notin C' \cup \{c_m\}$ with $i < j$ the alternative $c_i$ is ranked below $c_j$.*

That is, in $v$ the alternatives in $C'$ appear in the increasing order of indices, followed by $c_m$, followed by the remaining alternatives in the decreasing order of indices, i.e., the sequence of indices in $v$ is "single-peaked". Using this observation, we establish a bijection between the votes in the support of $\mathcal{D}_{\mathrm{GS}}^{\mathcal{T}}$ and single-peaked votes.

**Theorem 3.3.** *Given a ranking $v$ over $C = \{c_1, \ldots, c_m\}$, let $\widehat{v}$ be another ranking over $C$ such that, for each $j \in [m]$, if $c_j$ is ranked in position $i$ in $v$ then $c_i$ is ranked in position $m - j + 1$ in $\widehat{v}$. Suppose that $v$ is in the support of $\mathcal{D}_{\mathrm{GS}}^{\mathcal{T}}$, where $\mathcal{T} = \mathrm{CP}(c_1, \ldots, c_m)$. Then $\widehat{v}$ is single-peaked with respect to $c_1 \lhd \cdots \lhd c_m$.*

*Proof.* Suppose that $c_m$ is ranked in position $z$ in $v$. Then $c_z$ is ranked first in $\widehat{v}$.

Consider two candidates $c_x, c_y$ with $x < y < z$. We will prove that in $\widehat{v}$ candidate $c_y$ is ranked above $c_x$, i.e., $\mathrm{pos}_{\widehat{v}}(c_x) > \mathrm{pos}_{\widehat{v}}(c_y)$. Let $k = \mathrm{pos}_{\widehat{v}}(c_x)$, $\ell = \mathrm{pos}_{\widehat{v}}(c_y)$. Then, in $v$, alternative $c_{m-k+1}$ is ranked in position $x$ and alternative $c_{m-\ell+1}$ is ranked in position $y$. As we have $x < y < z$ and $v$ is sampled from $\mathcal{D}_{\mathrm{GS}}^{\mathcal{T}}$, by Lemma B.1 we have $m - k + 1 < m - \ell + 1$, and hence $k > \ell$.

Similarly, if we have two alternatives $c_x, c_y$ with $z < y < x$, we can show that $\mathrm{pos}_{\widehat{v}}(c_x) > \mathrm{pos}_{\widehat{v}}(c_y)$. Thus, $\widehat{v}$ is single-peaked with respect to $c_1 \lhd \cdots \lhd c_m$, as claimed. □

## C Missing Material From Section 3.4

Recall that our candidate set is $C = \{c_1, \ldots, c_m\}$ and the societal axis is $c_1 \lhd \cdots \lhd c_m$. We consider the Conitzer distribution. Let $f(i, j)$ be the probability that in a sampled vote candidates $c_i, \ldots, c_j$ appear in the top $j - i + 1$ positions. Next we find the values of $f(i, j)$ for all $i, j \in [m]$, and using them we establish the frequency matrix for the Conitzer distribution.

**Proposition C.1.** *Let $i, j$ be two integers with $1 < i \leq j < m$. Then $f(\ell, \ell) = 1/m$ for all $\ell \in [m]$, $f(1, m) = 1$, $f(i, j) = 1/m$, $f(1, i) = (i+1)/2m$, and $f(j, m) = (m-j+2)/2m$.*

*Proof.* The quantity $f(\ell, \ell)$ is simply the probability that $c_\ell$ is ranked first, so we have $f(\ell, \ell) = 1/m$ by the definition of the Conitzer distribution.

The equality $f(1, m) = 1$ is immediate from the definition of $f(i, j)$.

To show that $f(i, j) = 1/m$, we give a proof by induction on $j - i$. The base step holds because for each $\ell \in [m]$ we have $f(\ell, \ell) = 1/m$. Assume that for all integers $x, y$ such that $1 < x \leq y < m$ and $y - x < j - i$ we have $f(x, y) = 1/m$. The only way for candidates $c_i, \ldots, c_j$ to be ranked in top $j - i + 1$ positions under the Conitzer model is that, while generating the vote, we placed candidates $c_i, \ldots, c_{j-1}$ in top $j - i$ positions and then extended the vote with $c_j$ (the probability of this latter step is $1/2$), or we placed candidates $c_{i+1}, \ldots, c_j$ in top $j - i$ positions and then extended the vote with $c_i$ (again, the probability of the latter step is $1/2$). Thus:

$$f(i, j) = \tfrac{1}{2}f(i, j - 1) + \tfrac{1}{2}f(i + 1, j) = \tfrac{1}{2m} + \tfrac{1}{2m} = \tfrac{1}{m}.$$

This proves the claim for $f(i, j)$.

Next we show that $f(1, i) = (i+1)/2m$, using induction over $i < m$. For $i = 1$ we have $f(1, 1) = 1/m = 2/2m$, so our claim holds. Assume that it holds up to $i - 1$. There are only two ways in which candidates $c_1, \ldots, c_i$ can be ranked in the top $i$ positions: Either we first place $c_1, \ldots c_{i-1}$ in top $i - 1$ positions and then extend the vote with $c_i$ (the latter step has probability 1, since there is no candidate to the left of $c_1$), or we first place candidates $c_2, \ldots, c_i$ in the top $i - 1$ positions and then extend the vote with $c_1$ (the latter step has probability $1/2$, since the vote could also be extended with $c_{i+1}$). Thus, we have:

$$f(1, i) = f(1, i - 1) + \tfrac{1}{2}f(2, i) = \tfrac{i}{2m} + \tfrac{1}{2m} = \tfrac{i+1}{2m}.$$

This completes the proof for $f(1, i)$. The expression for $f(j, m)$ can be derived by symmetry: we have $f(j, m) = f(1, m - j + 1)$. □

**Theorem 3.5.** *Let $c_1 \lhd \cdots \lhd c_m$ be the societal axis, where $m$ is an even number, and let $v$ be a random vote sampled from $\mathcal{D}_{\mathrm{SP}}^{\mathrm{Con}}$ for this axis. For $j \in [\frac{m}{2}]$ and $i \in [m]$ we have:*

$$\mathbb{P}[\mathrm{pos}_v(c_j) = i] = \begin{cases} 2/2m & \text{if } i < j, \\ (j+1)/2m & \text{if } i = j, \\ 1/2m & \text{if } j < i < m - j + 1, \\ (m-j+1)/2m & \text{if } i = m - j + 1, \\ 0 & \text{if } i + j > m. \end{cases}$$

*Further, for each candidate $c_j \in C$ and each position $i \in [m]$ we have $\mathbb{P}[\mathrm{pos}_v(c_j) = i] = \mathbb{P}[\mathrm{pos}_v(c_{m-j+1}) = i]$.*

*Proof.* Consider a candidate $c_j$, $j \in [\frac{m}{2}]$, and a position $i \in [m]$. We proceed by case analysis:

1. If $i < j$, then there are two ways to generate a vote with $c_j$ in position $i$: Either candidates $c_{j+1}, \ldots, c_{i+j-1}$ are ranked in the top $i - 1$ positions or candidates $c_{j-i+1}, \ldots, c_{j-1}$ are ranked in the top $i - 1$ positions. In both cases, $c_j$ is ranked in the $i$-th position with probability $1/2$ (indeed, we have $i + j < m/2 + m/2 = m$, so in the former case both $c_j$ and $c_{i+j}$ could have been placed in position $i$, and also $j - i \geq 1$, so in the latter case both $c_j$ and $c_{j-i}$ could have been placed in position $i$). Thus we have:

$$\mathbb{P}[\mathrm{pos}_v(c_j) = i] = \tfrac{1}{2} \cdot f(j+1, i+j-1)$$
$$+ \tfrac{1}{2} \cdot f(j-i+1, j-1) = \tfrac{1}{2m} + \tfrac{1}{2m} = \tfrac{1}{m}.$$

2. If $i = j$, then either candidates $c_1, \ldots, c_{j-1}$ are ranked in the top $j - 1$ positions and the vote is extended with $c_j$ (with probability 1), or candidates $c_{j+1}, \ldots, c_{2j-1}$ are ranked in the top $j - 1$ positions and the vote is extended with $c_j$ (with probability $1/2$). Thus, we have:

$$\mathbb{P}[\mathrm{pos}_v(c_j) = j] = f(1, j-1) + \tfrac{1}{2} \cdot f(j+1, 2j-1)$$
$$= \tfrac{j}{2m} + \tfrac{1}{2m} = \tfrac{j+1}{2m}.$$

3. If $j < i < m - j + 1$ then there is only one possibility for $c_j$ to be ranked $i$-th: It must be that candidates $c_{j+1}, \ldots, c_{j+i-1}$ are ranked in the top $i - 1$ positions and the vote is extended with $c_j$ (which happens with probability $\frac{1}{2}$ because $j + i \leq m$).[3] Thus, we have:

$$\mathbb{P}[\mathrm{pos}_v(c_j) = i] = \tfrac{1}{2} \cdot f(j+1, i+j-1) = \tfrac{1}{2m}.$$

4. If $i = m - j + 1$, the analysis is similar to the case $i = j$: candidates $c_{j+1}, \ldots, c_m$ must be ranked in the top $m - j$ positions, in which case $c_j$ gets ranked in the $(m - j + 1)$-st position (with probability 1). Thus, we have:

$$\mathbb{P}[\mathrm{pos}_v(c_j) = m - j + 1] = f(j+1, m) = \frac{m - j + 1}{2m}.$$

5. If $i > m - j + 1$ then $\mathbb{P}[\mathrm{pos}_v(c_j) = i] = 0$, because both to the left of $c_j$ and to the right of $c_j$ there are fewer than $i - 1$ candidates.

The fact that for each candidate $c_j \in C$ and each position $i \in [m]$ we have $\mathbb{P}[\mathrm{pos}_v(c_j) = i] = \mathbb{P}[\mathrm{pos}_v(c_{m-j+1}) = i]$ follows directly from the symmetry of the Conitzer distribution and the fact that $m$ is even. $\qquad \square$

## D   Missing Material From Section 3.5

**Proposition D.1** (OEIS Foundation Inc. (2020))**.** *There is an algorithm that computes $S(m, k)$ using at most polynomially many operations.*

*Proof.* First, we note that for each $m' \in [m]$ we have $S(m', 0) = 1$. Further, for each $m' \in [m]$ and $k' \in [k]_0$ the following recursion holds:

$$S(m', k') = S(m', k' - 1) + S(m' - 1, k') - S(m' - 1, k' - m').$$

Using these two facts and standard dynamic programming, we can compute $S(m, k)$ using $O(mk)$ arithmetic operations. Since $k$ is at most $O(m^2)$, the running time is at most $O(m^3)$. $\qquad \square$

**Lemma D.2.** *There is an algorithm that computes $T(m, k, j, i)$ in polynomial time with respect to $m$.*

*Proof.* Our algorithm is based on dynamic programming. Fix some $m > 0$, $k \in [m(m-1)/2]$, and $j, i \in [m]$. We claim that:

$$T(m, k, m, i) = S(m - 1, k - (m - i)).$$

---

[3]It is impossible for the candidates from the left side of $c_j$ to take the top $i - 1$ positions because there are fewer than $i - 1$ of them.

Indeed, let $v$ be a vote over $C(m)$ that ranks $c_m$ in position $i$, and let $v'$ be its restriction to $\{c_1, \ldots, c_{m-1}\}$. Then $v$ can be obtained from $v'$ by inserting $c_m$ right behind the candidate in position $i - 1$. If $v'$ is at swap distance $k'$ from $v^*_{m-1}$, then the resulting vote is at swap distance $k' + (m - i)$ from $v^*_m$, since $c_m$ contributes $m - i$ additional swaps. Thus, $T(m, k, m, i)$ is equal to the number of votes in $\mathcal{L}(C(m - 1))$ at swap distance $k - (m - i)$ from $v^*_{m-1}$.

Next, we claim that for each $j < m$, we have:

$$T(m, k, j, i) = \sum_{\ell=i+1}^{m} T[m - 1, k - (m - \ell), j, i]$$
$$+ \sum_{\ell=1}^{i-1} T[m - 1, k - (m - \ell), j, i - 1].$$

To see why this holds, again consider inserting $c_m$ at some position in a vote $v'$ over $\{c_1, \ldots c_{m-1}\}$. Candidate $c_j$ will end up in position $i$ in the resulting vote if (1) $c_j$ was in position $i$ in $v'$ and $c_m$ was inserted after $c_j$, or if (2) $c_j$ was in position $i - 1$ in $v'$ and $c_m$ was inserted ahead of $c_j$. Considering all positions in which $c_m$ can be inserted, we obtain the above equality.

Using the above equalities and the fact that $T(1, 0, 1, 1) = 1$ (as there is just a single vote over $C(1)$), we can compute $T(m, k, j, i)$ by dynamic programming; our algorithm runs in polynomial time with respect to $m$. □

# E Missing Material From Section 4

## E.1 Distance to Compass Matrices

We analyze the distances between our matrices for different numbers of candidates. In Figure 5 we show these distances for the Conitzer and Walsh matrices and the compass matrices: For Conitzer, they are nearly constant, and for Walsh they vary significantly. Indeed, the more candidates we have, the closer the Walsh matrix is to ID (e.g., for 10 candidates their distance is 0.44, and for 300 candidates it is 0.09).

Figure 6 depicts the distance between the frequency matrix for GS/caterpillar and the four compass matrices, for varying number of candidates. As for the matrix for the Walsh model, its distance to the compass matrices changes as the number of candidates increases: The matrix moves closer and closer to AN.

In Figure 7, we display the distance between the frequency matrix for the Mallows model for different values of the dispersion parameter $\phi$ and the compass matrices (in contrast to the previous figures, we only consider up to 100 candidates, as for more than 100 candidates computing the matrix for the Mallows model becomes very memory-consuming). Independent of the chosen value of the dispersion parameter, the distance of the respective matrix to the four compass matrices changes significantly when we increase the number of candidates. In fact, for any fixed dispersion parameter $\phi$, the resulting matrix will always move closer and closer to ID as the number of candidates increases.

In contrast, if we use the normalized version of the Mallows model, the matrices remain more or less at a constant distance from the compass matrices. Figure 8 shows the distance of the frequency matrix of the normalized version of the Mallows model for different values of norm-$\phi$, as the number of candidates increases.

## E.2 Distances of Pairs of Vote Distributions on the Skeleton Map

As we have observed above, some vote distributions stay more or less at constant normalized positionwise distance from the four compass matrices. This raises the question whether these matrices also stay at a constant normalized positionwise distance from each other. This would imply that the data on which the skeleton map is based is independent of the number of candidates, and thus that the map is likely to look very similar for different numbers of candidates. To check this, we conducted the following experiment. We put together a set of vote distributions/matrices that do not structurally change when increasing the number of candidates (like the change happening for the Walsh model). First, we add the four compass matrices and the frequency matrix of the Conitzer model. Second, we add the frequency matrices of different variants of the Mallows model (similar as on the skeleton map as described in Section 4): the normalized Mallows model, the normalized Mallows model where the central vote is reversed with probability $1/2$, the normalized Mallows model

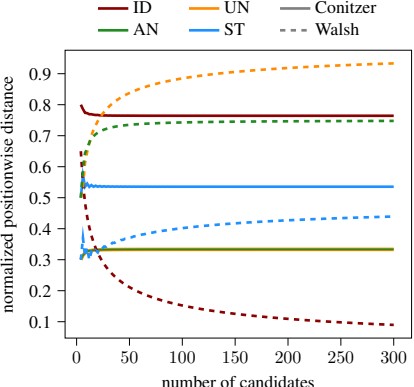

Figure 5: Normalized positionwise distance between the Conitzer [Walsh] matrix and the compass matrices in solid [dashed] lines, for varying number of candidates.

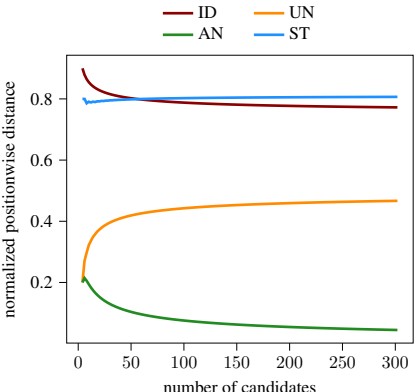

Figure 6: Normalized positionwise distance between the frequency matrix of the GS/caterpillar distribution and the compass matrices, for varying number of candidates.

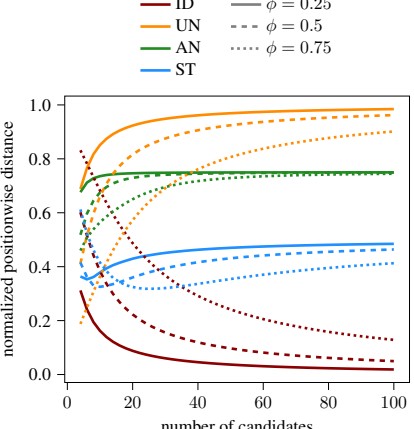

Figure 7: For different values of $\phi$, normalized positionwise distance between the frequency matrix of the Mallows distribution and the compass matrices, for varying number of candidates.

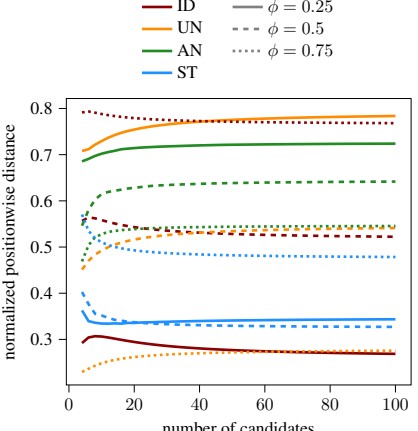

Figure 8: For different values of norm-$\phi$, normalized positionwise distance between the frequency matrix of the normalized Mallows distribution and the compass matrices, for varying number of candidates.

where the central vote is reversed with probability $1/4$, and the distribution where we first sample a vote $v$ from the Conitzer distribution and then sample the final vote from the normalized Mallows model with $v$ as the central vote. For each of these variants, we include their frequency matrix for norm-$\phi \in \{0.2, 0.4, 0.6, 0.8\}$.

For each pair of matrices from the created set, we compute their normalized positionwise distance for 100 candidates. Subsequently, for $m \in \{4, 6, \ldots, 98, 100\}$ candidates, we compute the normalized positionwise distance of the frequency matrices of the two considered models for this number of candidates as well as the absolute and relative difference between their normalized distance for $m$ and 100 candidates (where we normalize by their normalized distance for 100 candidates). Finally, for each $m \in \{4, 6, \ldots, 98, 100\}$, we take the maximum over the computed absolute/relative differences for all pairs of matrices. In Figure 9, we present these maxima for all considered values of $m$. Examining the maximum absolute difference (the blue line in Figure 9), what stands out is that for 20 or more candidates the normalized positionwise distance of any pair of considered vote distributions/matrices differs only by at most $0.0287$ from the pair's normalized positionwise distance for 100 candidates (for 10 or more candidates the error is at most $0.0547$). As the diameter of our

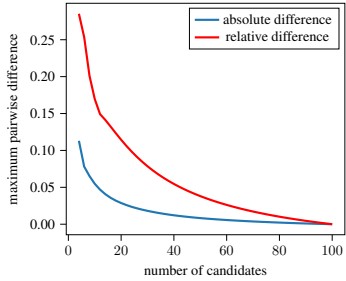

Figure 9: Results from our experiments described in Appendix E.2

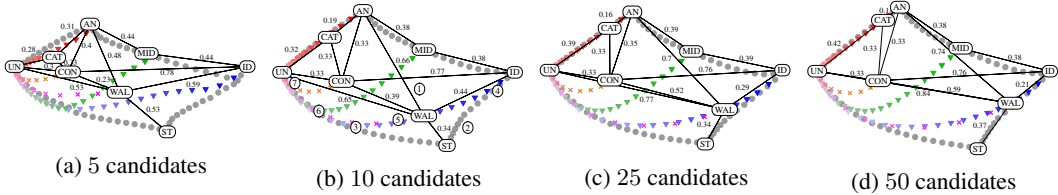

(a) 5 candidates      (b) 10 candidates      (c) 25 candidates      (d) 50 candidates

Figure 10: Skeleton map for different number of candidates.

space has at least length 1, this change is quite small, and the considered vote distributions indeed remain at nearly the same distance for more than 20 candidates. Considering the relative difference (the red line in Figure 9), the picture appears to be a bit worse: for 20 or more candidates, the normalized positionwise distance of any pair of considered vote distributions/matrices differs at most by 11.44% from their normalized positionwise distance for 100 candidates. Nevertheless, this value is still relatively low, indicating an overall high robustness of the normalized positionwise distances of each pair of considered distributions with respect to the number of candidates.

### E.3 Skeleton Map for Different Number of Candidates

After we have provided various arguments for why large parts of the skeleton map are presumably quite robust with respect to changing the number of candidates in the previous two subsections, in Figure 10, we present the skeleton map for 5/10/25/50 candidates. While the map for 5 candidates looks a bit different from the other maps, the maps for 10, 25, and 50 candidates differ only in that the frequency matrix for GS/caterpillar moves closer to AN and that the frequency matrix for the Walsh model moves closer to ID (both phenomena that we have already observed earlier).

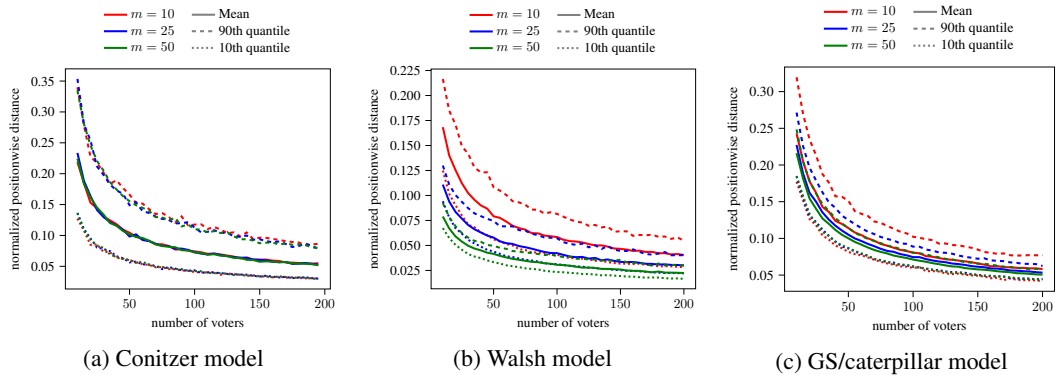

(a) Conitzer model      (b) Walsh model      (c) GS/caterpillar model

Figure 11: For different vote distributions, behavior of the normalized positionwise distance between elections sampled from this distributions and the distribution's frequency matrix, for 10/25/50 candidates and between 10 and 200 voters.

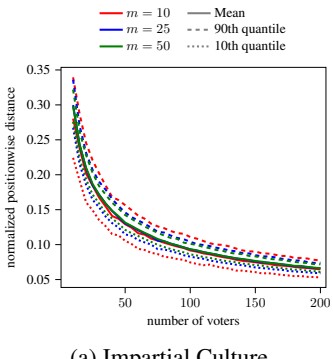

(a) Impartial Culture

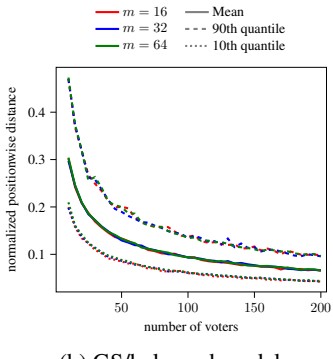

(b) GS/balanced model

Figure 12: For different vote distributions, behavior of the normalized positionwise distance between elections sampled from this distributions and the distribution's frequency matrix, for 10/25/50 candidates and between 10 and 200 voters.

### E.4  Variance of a Vote Distribution

As argued in Section 3.1, a frequency matrix of a distribution can be interpreted as a matrix of an "ideal" election sampled from this distribution. In this section we ask how far, on average, the elections sampled from our distributions land from the "ideal" ones. This distance may also serve as a measure of "diversity" for elections sampled from a given distribution.

For each of our distributions, we consider elections with 10, 25, and 50 candidates, and vary the number of voters from 10 to 200 (with a step of 5). For each combination of these parameters, we sample 600 elections and, for each election, compute the positionwise distance between its frequency matrix and the matrix of the respective distribution.[4] We show the results in Figures 11 and 12. As expected, for all vote distributions, increasing the number of votes decreases the average distance of an election from the distribution's matrix (indeed, in the limit this distance is zero). What is more surprising, this distance does not seem to depend on the number of candidates for the Conitzer, IC, and GS/balanced distribution. For the Walsh distribution (and, to a lesser extent, for GS/caterpillar), the sampled elections get slightly closer to the respective matrix as we increase the number of candidates. Moreover, if we fix the number of candidates and voters, then for all our distributions the elections sampled from them are, roughly, at the same distance from the distribution's matrix. For an illustration of this effect consider Figure 11a for the Conitzer distribution; there, the 10th quantile (dotted) and the 90th quantile (dashed) only differ by a factor of four. Lastly, comparing the plots for IC and GS/balanced (Figure 12), who both have UN as their frequency matrix, the average distance of elections sampled form one of these two models to UN (which is their frequency matrix) is the same for both distributions. However, for IC, the 10th and 90th quantile of the distances of elections to UN are closer to the average than for GS/balanced, which indicates that IC produces in some sense less varied elections than GS/balanced.

In Figure 13 we compare the average distances between elections sampled from various vote distributions and the distribution's matrices (we fix the number of candidates to 50 and vary the number of voters). While, on average, IC elections and GS/balanced elections end up at nearly the same distance from UN (which is their frequency matrix), Conitzer elections and GS/caterpillar elections end up closer to their distribution's matrices, and for Walsh this effect is considerably stronger. Overall, it is remarkable that even for 200 voters, for the Conitzer, IC, GS/balanced, and GS/caterpillar, the average distance of a sampled election from the respective matrix is still above $0.05$ (so at least $5\%$ of the diameter of the whole space). We also performed the same experiment for the Mallows model with different values of the normalized dispersion parameter (see Figure 14): For a varying number of voters, we depict the average distance of 600 elections with 50 candidates sampled from Mallows model for different values of norm-$\phi$ to the distribution's frequency matrix. Quite intuitively, the more swaps we make to the central vote (i.e., the higher norm-$\phi$ is), the higher is the average distance of a sampled elections from the distribution's frequency matrix.

---

[4]For the GS/balanced distribution we consider 16, 32, and 64 candidates, as this model requires the number of candidates to be a power of two and we do not consider GS/flat trees, as this distribution is too simple.

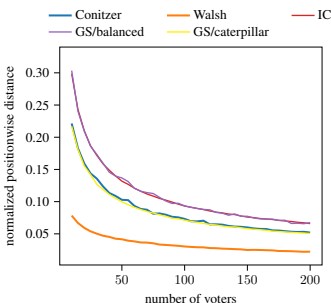

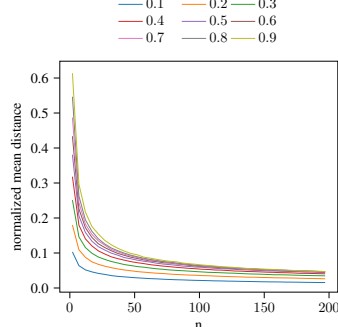

Figure 13: Average normalized positionwise distance between elections sampled from various voter distributions and the frequency matrices of the respective models, for 50 candidates (64 for GS) and between 10 and 200 voters.

Figure 14: For different values of norm-$\phi$, average normalized positionwise distance of elections with 50 candidates and between 10 and 200 voters sampled from the normalized Mallows model from the frequency matrix of the respective distribution.

It is interesting to contrast the data from Figure 11a with the map of Boehmer et al. (2021b) in Figure 2. Boehmer et al. considered 10 candidates and 100 voters. For these parameters, in Figure 11a we see that 10% of the elections are still farther from the Conitzer matrix than about 12% of the distance from UN to ID. This is roughly reflected by the size of the area taken by Conitzer elections in Figure 2. Similar observations hold for the other distributions too. While this might be a coincidence, it confirms the value of their map.

# F    Validation of Location Framework

Again, in this section, distances and dispersion parameters are always normalized. To validate whether our approach from Section 5 correctly identifies the "nature" of an election, we test its capabilities to find for a given election the dispersion parameter of the closest Mallows distribution with a single central vote. That is, given an election, we computed for all $\phi \in \{0, 0.001, \ldots, 1\}$ its distance to $\mathcal{D}_{\mathrm{Mal}}^{v,\phi}$ and returned the minimizing $\phi$ value. We compare the computed value to the maximum-likelihood estimator for the dispersion parameter of the underlying Mallows distribution computed from the Kemeny consensus ranking (Mandhani & Meilă, 2009) in two experiments.

We start by sampling for $\phi \in \{0, 0.05, \ldots, 1\}$ an election with 10 candidates and 100 voters from Mallows model with dispersion parameter $\phi$ and computed estimates for the dispersion parameter based on our and the Kemeny approach. For all elections, the returned estimates differ by at most 0.01. Thus, the dispersion parameter returned by our approach is always very close to the maximum-likelihood estimate. However, the estimated dispersion parameter might deviate a bit from the originally used dispersion parameter: On average, the absolute difference between the dispersion parameter returned by our approach and the underlying dispersion parameter is 0.0179 with the maximum difference being 0.069; for Kemeny, the average is 0.016 and the maximum is 0.082. While it might seem surprising that the estimated dispersion parameter is different from the underlying one, recall from Appendix E.4 that elections sampled from a vote distribution typically have non-zero distance from the distribution's frequency matrix. To illustrate this idea, we can think of a Mallows distribution as a normal distribution placed in the space of elections with the dispersion parameter being its mean (in particular as soon as the dispersion parameter is greater zero, all elections have a non-zero probability of being sampled). So multiple Mallows distributions for different dispersion parameters translate to multiple partly overlapping normal distributions and it might as well happen that an election sampled from a Mallows distribution with one dispersion parameter is in fact closer to the mean of the Mallows distribution with a different dispersion parameter.

We also repeated the above experiment to measure the capabilities of our approach to estimate the parameters of a mixed Mallows distribution. For each pair of $p \in \{0.1, 0.2, 0.3, 0.4, 0.5\}$ and $\phi \in \{0.05, 0.1, \ldots, 0.95\}$ we sampled an election with 10 candidates and 100 voters from $p\text{-}\mathcal{D}_{\mathrm{Mal}}^{v,\phi,\phi}$ (i.e., we sample a vote from the Mallows distribution with dispersion parameter $\phi$ and

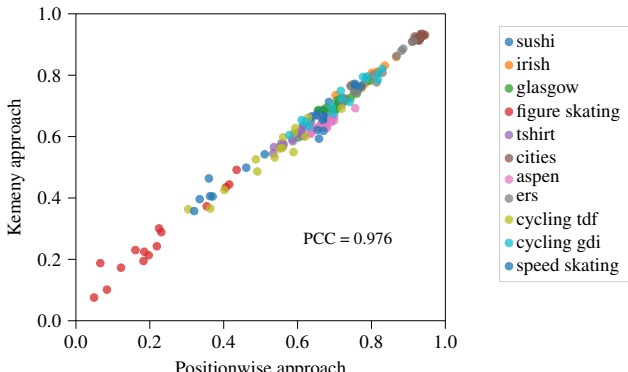

Figure 15: Correlation between the predicted dispersion parameter of our real-world data by our (positionwise) approach and by the maximum-likelihood Kemeny approach.

subsequently flip the sampled vote with probability $p$). Subsequently we computed for which value of $p \in \{0, 0.05, \ldots, 0.5\}$ and $\phi \in \{0, 0.05, \ldots, 1\}$ the distance of the sampled election is closest to the frequency matrix of the induced distribution $p\text{-}\mathcal{D}_{\text{Mal}}^{v,\phi,\phi}$ (the Kemeny consensus ranking can no longer be used here to provide a maximum-likelihood estimate). The average difference between the estimated and underlying dispersion parameter is $0.028$ and the average difference between the estimated and underlying flipping probability is $0.068$. For 63 out of the 95 elections is the difference between the estimated and underlying dispersion parameter and weight smaller equal $0.05$ (which is the smallest non-zero difference). The error of the estimated dispersion parameter here is around three times larger than for Mallows elections with a single central vote. This can be explained by the fact that for mixtures of Mallows distributions, the "overlap" between different distributions is even larger; in fact, some paramterizations even result in the same distributions (e.g., for $\phi = 1$ all flipping probabilities result in the same distribution).

Second, while producing good estimates for elections that have been sampled from a Mallows distribution is a good sanity check, we are ultimately interested in computing to which distribution (unknown) real-world elections are closest. To do so, we again compare the estimated dispersion parameter for a Mallows distribution with a single central vote computed by our approach with the one estimated via the Kemeny consensus ranking (as described in the beginning of this section); however, this time instead of considering elections sampled from Mallows model, we examine 165 real-world elections used by Boehmer et al. (2021b) (see the data part of Section 5 for details on the dataset). The estimated dispersion parameters returned by both methods are highly correlated with a Pearson correlation coefficient of $0.976$ and an average difference of $0.017$, median difference of $0.0105$, and maximum difference of $0.197$, indicating the power of our approach. Interestingly, the correlation is particularly strong for larger dispersion parameters (see Figure Figure 15 for a plot showing the correlation between the two approaches). Together with the estimated normalized dispersion parameter, both approaches also return the central order $v$ of the closest Mallows model, which are typically again quite similar: the average swap distance between the two estimators is $2.81$ out of 45 possible swaps.