# OpenReview forum: "Expected Frequency Matrices of Elections: Computation, Geometry, and Preference Learning"
_NeurIPS.cc/2022/Conference — NeurIPS 2022 Accept_

### Official Review · Reviewer_Y3YX · 2022-07-08

**Rating:** 5
**Confidence:** 3
**Soundness:** 3 good
**Presentation:** 4 excellent
**Contribution:** 2 fair

**Summary:**

Previously, Szufa et al. (2020) and Boehmer et al. (2021) proposed "map of elections" as both a visualization tool to group similar elections, but also demonstrated that those elections close to each other on the map may have similar features.This paper introduces "skeleton map", which is a new type of "map of elections". The main difference is that in the skeleton map, each point is a distribution instead of a single election. This change simplifies the map while preserving the key features. In order to represent each election / vote distribution as a point, this paper studies a number of prominent vote distributions, either gives an analytical formula of the frequency matrices, or provides a polynomial time algorithm to calculate it. The authors ran experiments to see how the distances between different vote distributions change when the number of candidates change. Finally, they use real life election datasets to verify that the vote distributions they consider (Mallows, Conitzer, Walsh with different parameters) could fit some real vote datasets well, but still have their limits.


**Questions:**

* On line 20: “However, performing high-quality experiments requires the ability to organize and understand the available data.” Could you elaborate on this? How the “map of elections” and “skeleton map” help experiments?
* For the frequency matrix calculation, I wonder if the results could be used in general above the context of this paper?
* Do you know whether your method in this paper is better than previous work? See my comments in “Strengths And Weaknesses”.


**Limitations:**

* It would be nice if the frequency matrix calculation could be used beyond the context in this paper.
* Again, my main concern is how this paper compares with previous papers on fitting real life datasets.


**Strengths And Weaknesses:**

I have previously reviewed another version of this paper.

* Originality: This work is well built on two previous works: Szufa et al. (2020) that proposed the "map of elections", and Boehmer et al. (2021) that defined compass matrices and normalized positionwise distance. They did a thorough work providing algorithms to compute the frequency matrices for many vote distributions. The "skeleton map" is a nice simplification to only have one point for each distribution instead of one point for each election.
* Quality: The paper is well organized. Concepts and algorithms are clearly defined, proved, and elaborated. Experiments and analysis of results are well presented. Source code is provided. My concern is that I did not find experiments showing this new method performs better than previous work.
* Clarity: This paper is well written and organized.
* Significance: My main concern is whether representing vote distributions by frequency matrices is better (in the sense that it fits better with real life datasets) than previous work that represents vote distributions by sampling elections from it.

---

> ### Author Response · Authors · 2022-08-01
> **Author Response**
>
> *Maps and experiments*
>
> The map shows relations between elections and between statistical cultures. As elections close on the map often have similar properties (see Boehmer et al. 2021a+b; Szufa et al. 2020), this is useful, e.g., to design diverse datasets, uniformly covering the space of elections. Further, by putting real-life elections on the map one can identify their nature (see, e.g., *Collecting, Classifying, Analyzing, and Using Real-World Elections*, by Boehmer and Schaar, available on arXiv). The skeleton map is particularly useful because it represents distances more accurately and relations between different models and real-life elections are clearer. Finally, the map provides a useful tool to visualize non-aggregated experimental results, which is an important issue in experimental sciences (we paint each election with a different color, according to the feature being visualized).
>
> &nbsp;
>
> *Further applications of frequency matrices*
>
> Expected frequency matrices offer a possibility to tackle different types of questions:
>
> 1) They enable one to reason about the asymptotic behavior of models. E.g., we observed that elections sampled from the Walsh model tend to the Identity election as the number of candidates increases. Using our results, one could compute the value to which the distance between Walsh and Identity matrices converges (and verify if it is 0). Similar asymptotic analysis for models whose distance seems to stay "constant" is also desirable. Sampling-based approaches cannot be used to prove such statements.
>
> 2) Exact frequency matrices allow us to reason about the nature of distributions.
> E.g., given the expected frequency matrix for the Walsh model, we see how often each candidate from the axis is ranked first. We could use this knowledge to (try to) answer the following question: If we chose the top candidate according to the probabilities as in the Walsh model, but then continued as in the Conitzer model (by extending the vote "to the left" and "to the right" with the same probability) would we obtain a model equivalent to that of Walsh? This would contribute to a better understanding of the Walsh model.
>
> &nbsp;
>
> *Comparison with previous work*
>
> In Appendix G, we compare our approach for learning the dispersion parameter of Mallows model to the maximum-likelihood estimator (based on the Kemeny ranking). Recall that the dispersion parameter lies between $0$ and $1$. We found both approaches to be highly correlated (average absolute error $0.017$, median absolute error $0.01$, mean squared error $0.0008$, PCC $0.976$). However, while the Kemeny ranking is NP-hard to compute, our approach reduces to computing a maximum-weight matching. Methods from previous works often reduce to the NP-hard Kemeny estimator for a single Mallows distribution.
>
> &nbsp;
>
> *Is representing vote distributions by frequency matrices better than representing vote distributions by sampling elections*
>
> To fit a real-life election $E$ one can also create for each vote distribution $D$ in question an election $E_D$ with sufficiently many votes and then compute the distance between $E$ and $E_D$. This approach boils down to approximating the frequency matrix of $D$ by the frequency matrix of $E_D$ instead of relying on our exact algorithms and formulas. However, it has disadvantages: First, all our distributions have high variance, i.e., sampled votes can be quite distant from one another (see Appendix F.4). Second, sampling votes is time-consuming. E.g., for 10 candidates, computing the expected frequency matrix using our approach takes as long as sampling 1 vote, 2 votes, and 20 votes from Conitzer, Walsh, and Mallows models, respectively (for the last one, we used parameter 0.5). If we estimate an expected frequency matrix $\hat{M}$ via sampling, giving it 10 times more time than needed for the respective exact matrix $M$,  the summed absolute error: $$\sum_{i,j\in [10]} |M_{ij}-\hat{M}_{ij}|$$ is 8.4, 3.2, and 1.39 for Conitzer, Walsh, and Mallows 0.5 models, respectively (for 10 candidates). Recall that both in $M$ and $\hat{M}$ all entries sum up to 10. For a factor of 100, we get an error of 1.88, 1.37, and 0.49, and for a factor of 10000, we get 0.2, 0.12, and 0.047. Thus, even if we sample many votes, we still have a considerable error, which affects further results. E.g., we rerun our experiments on learning the dispersion parameter (see Section 5, starting on line 365) but instead of using the exact frequency matrices, we used for each dispersion value an election with 200 sampled votes; this experiment took roughly 10 times longer than the original one and the results got notably worse. Comparing the estimated dispersion parameter to the Kemeny ML-estimator for the dispersion parameter, the average absolute error between the two is now 0.1 (instead of 0.017), the median error is 0.05 (instead of 0.01), the mean squared error is 0.026 (instead of 0.0008), and the PCC is 0.76 (instead of 0.976).

---

> > ### Comment · Reviewer_Y3YX · 2022-08-09
> > **Thank you for the response!**
> >
> > Thank you for the response and clarification! I have updated my review score.

---

### Official Review · Reviewer_m4hd · 2022-07-11

**Rating:** 7
**Confidence:** 3
**Soundness:** 3 good
**Presentation:** 3 good
**Contribution:** 4 excellent

**Summary:**

The authors build on recent work on a very exciting "map of elections" by showing how to compute the frequency matrix of various vote distributions. They then use these frequency matrices to produce a "skeleton map" of distributions that is closely related to, and visually much simpler than, the "map of elections" approach. Lastly, they show that they can use frequency matrices to estimate parameters of real-world election data (based on nearest distributions).


**Questions:**

1. Line 22: what is eu?

**Limitations:**

Yes

**Strengths And Weaknesses:**

Strengths:
+ The "map of elections" is a remarkably useful and insightful tool when dealing with real-world elections, and having a "skeleton map" like this (that doesn't rely on sampling individual instances) is a very useful addition.
+ I particularly appreciated that this skeleton map allows us to learn parameters of real-world elections (or approximations thereof).
+ The sampling results / algorithms are intuitively presented and clear to the reader (despite the fact that the results are quite nontrivial).

Weaknesses:
- I am not overly familiar with the background behind the map of elections (beyond seeing it as a very useful tool), and it seems like the whole area is built upon many heuristics that seem to work pretty well but lack robust theoretical justification. This is perhaps not so much of a downside to me (relative to others) but is definitely something worth thinking about.

---

> ### Author Response · Authors · 2022-08-01
> **Author Response**
>
> *I am not overly familiar with the background behind the map of elections (beyond seeing it as a very useful tool), and it seems like the whole area is built upon many heuristics that seem to work pretty well but lack robust theoretical justification. This is perhaps not so much of a downside to me (relative to others) but is definitely something worth thinking about.*
>
>
> The map framework relies on two main heuristics: the embedding algorithm, for computing the visualizations, and the positionwise distance, for analyzing similarities between elections and distributions.
> Regarding the embedding heuristic, Figure 4 shows that our plots are quite credible, especially due to using the skeleton map, which includes substantially fewer points than the previous maps.
> However, indeed, we cannot offer theoretical explanations as to why this is the case (except for the metaargument that we use well-known embedding techniques). Regarding the positionwise distance, we refer to the following
> recent paper:
>
> &nbsp;&nbsp;&nbsp;&nbsp;&nbsp;&nbsp;Niclas Boehmer, Piotr Faliszewski, Rolf Niedermeier, Stanislaw Szufa, Tomasz Was: Understanding Distance Measures Among Elections. IJCAI-2022.
>
> The authors argue that the isomorphic swap distance would be the ideal distance to use in the map framework, but it is computationally far too expensive (they report that computing a map of elections similar to the one from Figure 2 for 10 candidates and 50 voters took them a few weeks). Then they provide various experimental and theoretical arguments showing that among several other distances, the EMD-positionwise distance---used in the original map papers, as well as in our work---seems very good both in terms of its expressivity and in terms of its computational complexity.
>
> &nbsp;
>
> *Line 22: what is eu?*
>
> This is a typo, which escaped our quality check (analysis of the paper's edit history shows that it was created accidentally right before final compilation and submission). This sentence should have read:
> "One way to achieve this is to form a so-called `map of elections,' recently introduced
> by Szufa et al. (2020) and extended by Boehmer et al. (2021b)."

---

> > ### Comment · Reviewer_m4hd · 2022-08-09
> > **Author response**
> >
> > Thank you for your comments!

---

### Official Review · Reviewer_vnFh · 2022-07-11

**Rating:** 8
**Confidence:** 4
**Soundness:** 4 excellent
**Presentation:** 3 good
**Contribution:** 4 excellent

**Summary:**

Note: I have reviewed a previous submission of this paper at a previous conference.

This paper continues a recent line of work aimed at identifying relationships between many different vote distributions. Where previous work used frequency matrices to measure the difference between sampled elections from given distributions this paper introduces a very natural extension to the concept and applies the frequency matrices to the vote distributions themselves.

The paper describes a number of election structures (single-peaked, group-seperable, Mallows) and finds the frequency matrix, or a polynomial time formula for generating the matrix, for each distribution. Using these frequency matrices a new map of elections is generated. This "skeleton map" strengthens and confirms prior work and is shown to generally represent the distance between elections well. Finally, Mallows models are generated for elections representing real-world data and placed on the map. These also have some similarity to prior work however they find generating a distribution to perfectly match each real election to be difficult. Some promising potential future work is identified while concluding.

**Questions:**

Can you discuss why this paper was not previously accepted and how it has changed since then?

Tiny issue:
Line 290 misspells "Conitzer" as "Cointzer"

**Limitations:**

The authors have not discussed the potential societal impact of their work. While the contribution appears quite far removed from any possible real-world impact perhaps one of the many appendices could be used to briefly imagine how this work could be misused somewhere down the line?

**Strengths And Weaknesses:**

The paper represents a novel addition to a recent series of papers. This approach of generalizing from sampled data to comparing entire distributions could also have potential uses in other domains. The work is well written and does a very good job of connecting itself to the prior results in this line of research. While the results are moderately complex I found them to be explained and structured quite clearly.

Overall I find the paper fairly strong and have no major issues with it. Due to space limitations some of the figures (particularly Fig 2) are rather small and difficult to read. I am glad to see that most of the specific, minor changes I have previously suggested have been fixed.

---

> ### Author Response · Authors · 2022-08-01
> **Author Response**
>
> *Can you discuss why this paper was not previously accepted and how it has changed since then?*
>
>
> Our paper was rejected from the previous venue with the following meta-review:
>
> "This paper looks at properties of election systems, following the line of work of Szufa et al.
>
> The paper is extremely clear and well written, and I enjoyed reading it. This said, I am very concerned about the relevancy to the ICML community. Admittedly, and I totally agree with this, it would/should be a very interesting paper for a large chunk of the community, but in its current form, it misses the theoretical and/or practical contributions to reach the global bar.
>
> As a consequence, I must unfortunately recommend rejection (and I would suggest an alternative venue, more focused on computational social sciences)."
>
> &nbsp;
>
>
> We have addressed this criticism as follows:
> 1. We have revised substantial parts of the introduction and the experimental section of the paper in order to position our work more adequately within the literature and to show the applicability of our results to preference learning.
> 2. While doing the above, we have realized that we have missed a substantial chunk of related literature published in machine learning venues such as JMLR, NeurIPS, and ICML, which we now reference and discuss.
> 3. We decided to submit to NeurIPS as it appears to be broader in scope than ICML. In particular, the NeurIPS call for papers explicitly lists algorithmic game theory among its subject areas, and our paper belongs to the field of computational social choice, which is usually viewed as a subfield of algorithmic game theory.

---

### Meta-Review · Area_Chair_uGbU · 2022-08-30

**Recommendation:** Accept
**Confidence:** Certain

**Metareview:**

This paper works to identify relationships among different vote distributions. This is done by applying (previously introduced) "frequency matrices" to the vote distributions themselves and gives formula or algorithms for computing these.  The resulting "map of elections" seems to have especially strong real-world potential.

**Award:**

No

---

### Decision · Program_Chairs · 2022-09-14

Accept